# Molecular Phylogeny and Species Delimiting for the Genus *Hoplolaimus* (Nematoda: Tylenchida) with Description of *Hoplolaimus floridensis* sp. n. and Notes on Biogeography of the Genus in the United States

**DOI:** 10.3390/ijms26178501

**Published:** 2025-09-01

**Authors:** Sergei A. Subbotin, Mihail Kantor, Erika Consoli, Niclas H. Lyndby, Amy Michaud, Zafar Handoo, Renato N. Inserra

**Affiliations:** 1Plant Pest Diagnostic Center, California Department of Food and Agriculture, 3294 Meadowview Road, Sacramento, CA 95832, USA; amy.michaud@cdfa.ca.gov; 2Department of Entomology and Nematology, University of California, Davis, CA 95616, USA; 3Center of Parasitology of A.N. Severtsov Institute of Ecology and Evolution of the Russian Academy of Sciences, Leninskii Prospect 33, Moscow 117071, Russia; 4Plant Pathology & Environmental Microbiology Department, The Pennsylvania State University, University Park, PA 16802, USA; mpk6148@psu.edu (M.K.); consoli@psu.edu (E.C.); nfl5281@psu.edu (N.H.L.); 5Mycology and Nematology Genetic Diversity and Biology Laboratory, USDA, ARS, Northeast Area, Beltsville, MD 20705, USA; zhandoo@gmail.com; 6Florida Department of Agriculture and Consumer Services, DPI, Nematology Section, P.O. Box 147100, Gainesville, FL 32614, USA; renato.inserra@fdacs.gov

**Keywords:** D2–D3 of 28S rRNA gene, *COI* gene, ITS rRNA gene, lance nematodes, morphology, taxonomy

## Abstract

Lance nematodes, *Hoplolaimus* spp., feed on the roots of many kinds of plants, including agronomic crops. In this study, morphological and molecular analyses of several *Hoplolaimus* species and populations are provided. We were able to collect and characterize the topotype materials of *H. galeatus* from Arlington, Virginia; *H. stephanus* syn. n. from Nichols, South Carolina; and *H. concaudajuvencus* from Pensacola, Florida, and several additional populations and species from the United States, Israel, and India. Phylogenetic analyses of several hundred sequences of the D2–D3 expansion regions of 28S rRNA, ITS rRNA, and COI genes of *Hoplolaimus* species obtained from published and original datasets were given. Fifty-three new D2–D3 of 28S rRNA, 43 new ITS rRNA, and 47 new *COI* sequences from 23 isolates of *Hoplolaimus* spp. and one isolate of *Peltamigratus christiei* were obtained in this study. New molecular identities for *H. concaudajuvencus* and *H. galeatus* were proposed. *Hoplolaimus stephanus* syn. n. was considered a synonym of *H. galeatus* based on the morphological and molecular similarity of these two species. Analysis of morphology and molecular data did not reveal significant differences among *H. columbus* syn. n., *H. indicus* syn. n., and *H. seinhorsti*, and the first two species were synonymized with *H. seinhorsti*. A new species, *H. floridensis* sp. n., was described from many locations in Florida, USA. It was separated from other representatives of the genus *Hoplolaimus* by its morphological and molecular characteristics. Maps with geographical distribution of several lance nematode species in North America were reconstructed based on published and original molecular identification of samples.

## 1. Introduction

The genus *Hoplolaimus* Daday 1905 includes nematodes parasitizing a wide range of plants across the globe [1,2,3,4,5]. Members of this genus are commonly known as ‘lance nematodes’ because of their very well-developed and robust stylet having distinctly tulip-shaped knobs. They also have a distinct cephalic region, a massive cephalic framework, and phasmids that are normally not opposite one another. The lance nematodes are considered ectoparasitic, although they can act as semi- or facultative ecto-endoparasites. The type species of this taxon is *Hoplolaimus tylenchiformis* Daday 1905, described from an island in the Paraguay River [6]. According to Ghaderi et al. [3], the genus comprised 37 valid species, which can be identified using the diagnostic compendia provided by these authors and Handoo and Golden [7].

Fortuner [8] proposed separating *Hoplolaimus* species morphologically into four groups, defined by the presence of ancestral or derived characters. He considered that the ancestral characters included three gland nuclei and the position of the excretory pore below the hemizonind and lateral lines marked by four incisures, whereas the derived characters included six gland nuclei and the position of the excretory pore above the hemizonid and lateral field marked by one or no incisure. Two new genera were proposed to accommodate some *Hoplolaimus*: *Hoplolaimoides* by Shakil [9] and *Basirolaimus* by Shamsi [10]. The genus *Hoplolaimoides* included only one species, *H. californicus* Sher 1963, which was distinct from other *Hoplolaimus* species for having both scutella posterior to the vulva, but later this genus was invalidated by Siddiqi [11]. The genus *Basirolaimus* was established for nine *Hoplolaimus* species having six pharyngeal nuclei, including *H. seinhorsti* Luc 1958, *H. columbus* Sher 1963, and *H. indicus* Sher 1963, but subsequently Luc [12] proposed this genus as a junior synonym of *Hoplolaimus*. Siddiqi [13] distinguished three groups or subgenera for *Hoplolaimus* species originating in different regions: (i) *Hoplolaimus* containing type *H. tylenchiformis* and nine other species with tiled head bearing a round perioral disc, lateral fields with four incisures up to tail tip, excretory pore close to hemizonid and uninucleate dorsal pharyngeal gland, originating in South America and spreading to North America; (ii) *Basirolaimus* containing type *H. seinhorsti* and other seventeen species, having head with irregular cuticular divisions and bearing a lemon-shaped perioral disc, lateral fields obliterated, excretory pore at a distance from the hemizonid and dorsal pharyngeal gland with four nuclei, originating in the Oriental Region; and (iii) *Ethiolaimus* Siddiqi 2000 containing type *H. pararobustus* (Schuurmans Stekhoven & Teunissen 1938) Sher 1963 and other three species with characters similar to *Basirolaimus* but having a roughly rectangular perioral disc and uninucleate dorsal pharyngeal gland, originating in and being largely confined to Africa.

In the United States, nine *Hoplolaimus* species were reported: *H. californicus* Sher 1963, *H. columbus*, *H. concaudajuvencus* Golden & Minton 1970, *H. galeatus* (Cobb 1913) Thorne 1935, *H. magnistylus* Robbins 1982, *H. seinhorsti*, *H. smokyensis* Max, Robbins, Bernard, Holguin & Audelo 2019, *H. stephanus* Sher 1963, and *H. tylenchiformis. Hoplolaimus galeatus* was considered the most widely distributed species in the country. A species morphologically similar to *H. galeatus* is *H. stephanus,* originally described from Nichols, South Carolina, from swamp soil [3,7]. *Hoplolaimus stephanus* and *H. tylenchiformis* differed from *H. galeatus* by minor morphological characters, and reliable identification and separation of these three species are difficult. Except for *H. columbus,* which is closely related to *H. seinhorsti*, the remaining species reported in the United States are well defined and distinct morphologically.

A correct identification of these species is important for regulatory purposes and to improve our knowledge concerning the diversity of plant-parasitic nematode species infesting USA agricultural crops. The clarification of the taxonomic status of the *Hoplolaimus* species requires morphological and molecular comparisons using topotype populations of these species and DNA sequences available in the literature [4,5,14,15,16,17,18,19,20,21,22,23,24,25,26].

In the beginning of our study, we carried out phylogenetic analyses using DNA sequences of samples preliminarily identified as *H. galeatus* from Florida, a topotype population of this species from Virginia, and *H. stephanus* reported in the literature and deposited in the GenBank. The results of these analyses cast doubts about the identity of some previously studied lance nematode species. The disagreement in the results of this preliminary analysis with those reported in the literature prompted us to carry out a study to clarify the identity of these and other populations of lance nematodes in the United States and other countries.

In this study, we were able to obtain the topotype specimens of *H. galeatus* from Arlington, Virginia; *H. stephanus* from Nichols, South Carolina; and *H. concaudajuvencus* from Pensacola, Florida. Several additional populations and species of lance nematodes were collected in the United States: California, Delaware, Florida, Iowa, Maryland, Minnesota, Michigan, North Dakota, Pennsylvania, and South Carolina, and in two other countries: Israel and India. DNA sequences of lance nematodes from additional countries reported in the literature and deposited in the GenBank were included in these analyses.

The main aim of this study was to re-evaluate the species biodiversity of the genus *Hoplolaimus* in the United States. The objectives included: (i) morphologically characterize topotype populations of *H. galeatus*, *H. stephanus*, and *H. concaudajuvencus*; (ii) molecularly characterize topotype populations of these three species and also newly obtained populations and species of *Hoplolaimus* using sequences of the D2–D3 expansion segments of the 28S rRNA, the ITS rRNA, and partial *COI* mtDNA genes; (iii) revise previously published molecular identification of some *Hoplolaimus* species; (iv) describe a new species, *H. floridensis* sp. n. from Florida; (v) reconstruct phylogenetic relationships within *Hoplolaimus* species using rRNA and *COI* gene sequences and (vi) reconstruct maps of geographical distribution of some *Hoplolaimus* species based on published and original molecular identification of samples from the United States.

## 2. Results

In the results of morphological and molecular analyses of original and published datasets, we distinguished several *Hoplolaimus* species in the United States. Some of these species, such as *H. concaudajuvencus*, *H. galeatus*, *H. magnistylus*, *H. smokyensis*, and *H. seinhorsti,* were already reported in the literature; another species, such as *H. floridensis* sp. n., is described herein as a new species, and several others, such as *Hoplolaimus* spp. (A, B1, B2, and 1), remain unidentified (Table 1). Based on the results of characterization of the type populations using nuclear and mitochondrial genes, new molecular identities for *H. galeatus* and *H. concaudajuvencus* are proposed. As shown by the results of the phylogenetic analyses in the following sections of this paper, the molecular characters of the populations of lance nematodes from South Carolina, identified as *H. stephanus,* were congruous with those of a topotype population of *H. galeatus,* suggesting that *H. stephanus* syn. n. should be considered a junior synonym of *H. galeatus*. In contrast, the DNA sequences of lance nematodes from Florida, incorrectly identified as *H. galeatus*, did not group in the same clades as the topotype and other populations of this species. They represented a new species of *Hoplolaimus* that is described herein with the name of *H. floridensis* sp. n. *Peltamigratus christiei* (Golden & Taylor 1956) Sher 1963 from Florida was molecularly characterized for the first time.

### 2.1. Hoplolaimus floridensis sp. n. (Figure 1, Figure 2, Figure A1, Figure A2I and Figure A4D,I,J; Table A1)

=*Hoplolaimus neocoronatus* Whitten 1957 nomen nudum

Adult female. Body cylindroid, vermiform, slightly curved to open C-shaped after fixation, tapering slightly at both ends. Head with massive cephalic framework, set off from the rest of the body by a deep constriction with seven annuli, with the basal one subdivided into about 20–24 plate-like blocks/longitudinal striations. The head patterns were consistent with those described by Geraert [27] for species of the family *Hoplolaimidae*. Some modifications, however, were observed. *En face* views of the labial plate showed a prominent round oral disc with a round stoma in the center. Two small and pocket-like amphidial apertures are delimited by the oral disc and the small lateral lip sectors, which are fragmented into two or three small pieces. The sub-ventral and sub-dorsal lip sectors are prominent and trapezoidal, and some of them are divided into unequal parts. The lip annuli beneath the labial plate are incomplete and fragmented in various subdivisions. Blocks in the basal annulus are also irregularly subdivided into small and superimposed units. The incomplete lip annuli and the subdivided blocks of the basal annulus complicated the count of the lip annuli at LM, which can be seven on one side and five or six on the other side of the head. The stylet is strong with tulip-shaped knobs. Pharyngeal glands overlapping the intestine dorsally with three gland nuclei. The excretory pore is near the level of or posterior to the pharyngeal–intestinal valve; in a few specimens, it is located at about the posterior third of the pharyngeal glands. The hemizonid is mostly inconspicuous, only seen in a few specimens at about 3–4 annuli anterior to the excretory pore. The hemizonion is at about 7 annuli posterior to the excretory pore. Lateral field with four prominent lines, strongly aerolated throughout the body. Cuticular annulations are prominent with smooth annules. Phasmids (scutella) are large, conspicuous, and variable in position, with one at the anterior half and the other at the posterior half of the body. The vulva is a deep and prominent transverse slit near a little over mid-body; the ovaries are two and outstretched. Spermatheca round to oval with rounded sperm. The tail is broadly truncate to occasionally rounded to sort of clavate-shaped, noted in a few specimens with 7–14 annuli, and the c’ ratio is small, 0.6 ± 0.1 (0.5–0.8).

Adult male. Similar to female, except for sexual dimorphism. Basal annulus of head region with 20–24 longitudinal striations. Spicules, gubernaculum, and bursa are large and prominent, with the bursa extending to the tail tip and the c’ ratio greater than that of females at 1.5 ± 0.1 (1.5–1.7). There is one testis outstretched anteriorly.

Type host and locality. Associated with roots and soil around Bermuda grass (*Cynodon dactylon* Pers.) in Citra, Marion County, Florida, USA. The global positioning coordinates for Citra: latitude—29°40′82.84″ N and longitude—82°16′75.51″ W.

Other localities and hosts. Associated with roots and soil around St. Augustine grass (*Stenotaphrum secundatum* (Walt.) Kuntze) in Gainesville, Alachua County, Florida, USA. The global positioning coordinates for Gainesville: latitude—29°66′75.38″ N and longitude—82°38′95.80″ W. This species was also found in several locations in Collier County and Hendry County (Table 1).

Type material. The holotype (1 female: Slide T-831t) and paratype females and males are deposited in the United States Department of Agriculture Nematode Collection, Beltsville, MD, USA. Slides T-8270p-T-8272p for female specimens and slides T-8273p-T8277 for male specimens, as well as other paratypes, are in the Florida Department of Agriculture and Consumer Services, DPI, Nematology Section, Gainesville, Florida, USA. ZooBank urn: lsid:zoobank.org:pub:B8CDF251-AB33-418B-A7D7-FBC323459E6E.

Etymology. The specific epithet *floridensis* is derived from the Latin adjectivation of the name Florida, the state in the United States where the new species was recovered from five locations, including Citra and Gainesville.

Diagnosis and Relationships. *Hoplolaimus floridensis* sp. n. is characterized by a combination of the following morphological features in females: body is cylindroid, vermiform, and slightly curved to open C-shaped after fixation. The lip region is set off from the body contour, with five-seven annuli divided into sub-dorsal, sub-ventral, and lateral lip sectors (SEM); the stylet is strong with tulip-shaped knobs; pharyngeal glands are with three nuclei; the vulva is located at 51–59% of body length; the spermatheca is round to oval-shaped with rounded sperm; the tail is broadly truncate to occasionally rounded to sort of clavate-shaped with 7–14 annuli; and in males, the spicules, gubernaculum, and bursa are large and prominent, and there is one testis outstretched anteriorly. We would like to point out that head patterns are variable in some *Hoplolaimus* species [28]. In our examination at SEM, divided lip sectors were observed in all the specimens of *H. floridensis* sp. n. fixed for SEM observations. Additional SEM observations of populations of this species are needed to confirm or disprove the stability of this diagnostic character.

*Hoplolaimus floridensis* sp. n. is morphologically and morphometrically closely related to *H. galeatus*, *H. tylenchiformis*, and *H. stephanus*. It differs from the most closely described species, *H. galeatus* by having a slightly greater mean stylet length of 46.7.µm in Citra, respectively, 47.7 µm in Gainesville population vs. 45.0 µm with the Virginia-type location specimens except for Sher [29], where only range is given as 43.0–52.0 µm, head annuli seven with basal annulus having 20–24 longitudinal striations vs. five lip annuli with basal annulus having 32–36 longitudinal striations, epiptygma is absent or inconspicuous vs. single or double, usually conspicuous, lateral field with four incisures like blocks, heavily aerolated throughout body vs. four lightly and not completely aerolated incisures, anterior end to the base of pharyngeal bulb distance is longer 150.0–207.0 µm vs. shorter distance 100.0–175.0 µm, intestine not overlapping rectum vs. intestine overlapping rectum, tail broadly truncate to occasionally rounded sort of clavate appearance in few specimens with 7–14 annuli vs. tail rounded with 9–12 annuli; in males mean spicule length is greater 49.0 µm ± 5.1 (41.0–60.0 µm) vs. slightly shorter 44.0 µm ± 1.7 (42.0–45.0 µm). From *H. tylenchiformis,* the new species differs in having a longer body length in both females and males 1385.0- 1827.0 µm and 1305.0–1893.0 µm in Gainesville and Citra female populations, and 1150.0–1545.0 µm and 1050.0–1400.0 µm in male Gainesville and Citra populations vs. the shorter body length, 800.0–1300.0 µm and 700.0–1100.0 µm in females and males, respectively, of *H. tylenchiformis;* the number of lip annuli is greater, seven vs. the smaller three-four annuli; the intestine does not overlap the rectum vs. overlapping rectum; the epiptygma is absent vs. usually single and inconspicuous; and the tail is truncate vs. bluntly rounded. In males, spicules and gubernaculum are slightly longer, 41.0–60.0 µm and 21.0–32.0 µm, vs. short spicules and gubernaculum of 38.6 µm and 19.0 µm, respectively. From *H. stephanus*, the new species differs by having a greater number of lip annuli, seven vs. the smaller four annuli; intestine not overlapping rectum vs. overlapping rectum; and tail truncate vs. rounded. In males, spicules and gubernaculum are longer at 41.0–60.0 µm and 21.0–32.0 µm vs. shorter spicules and gubernaculum of 34.0 µm and 18.0 µm, respectively. *Hoplolaimus floridensis* sp. n. shares many morphological characters, such as the number of lip annuli (6–7) and incisures (4) in the lateral fields, with *H. californicus* and *H. igualaensis* Cid del Prado 1994, but it differs from these two species in having phasmids above and below the center of the body vs. below the center of the body and different patterns in the arrangement of the areolations in the lateral fields.

### 2.2. Hoplolaimus galeatus (Figure A2A–G,I, Figure A3A–G,K and Figure A4A–C,G–I; Table A2)

In our study, we compared the morphology and morphometrics of the topotype population that we collected with those reported by Sher [29] for a population from the type locality in Arlington, Virginia, for this species and other locations.

Adult female. Females of the topotype population that we collected showed morphological characters that were congruous with those reported by Sher [29] for this species. They had a cylindroid body and a lip region with five lip annuli, a length of stylet in the range of that reported by Sher (43.0–52.0 µm) with anteriorly dentate tulip-shaped knobs, and a pharyngeal gland lobe with three nuclei. Their hemizonid was located two annuli anterior to the excretory pore. The lateral field was aerolated and marked by four incisures. The spermatheca is conspicuous and filled with sperm. The vulva is like a transverse, deep slit. Pre-vulvar and post-vulvar phasmids are present. The tail is rounded to bluntly rounded.

Adult male and second-stage juvenile. The morphological characters of males and second-stage juveniles were consistent with those in the original description.

Remarks. Although morphometric characters of the topotype population from Virginia fell within the range of those reported by Sher [29], there were some slight differences in some characters, such as the distance between the anterior end and the pharyngeal gland lobe and the gubernaculum length, which were, respectively, shorter and longer in the population described by Sher [29] than in our topotype population. The number of five lip annuli in the cephalic region is a valid diagnostic character for the identification of this species.

### 2.3. Hoplolaimus galeatus from Nichols (Figure A2K and Figure A3K; Table A3)

*=Hoplolaimus stephanus* syn. n.

The topotype population of *H. stephanus* has remained elusive for decades, preventing the molecular characterization of this species. The morphological description by Sher [29] has been the only source of information about the characteristics of this species, which has a lip region with four annuli. Examination of *H. stephanus* paratypes deposited at the USDA Nematode Repository provided evidence of variability in the number of lip annuli of the examined specimens, which exhibit five lip annuli rather than four as reported in the original description. In April 2025, we were able to collect for the first time topotype specimens of *H. stephanus* from Nichols, South Carolina. The morphometrical results are presented in Table A3, and the molecular characters of this topotype population are reported in the following sections. The Nichols population was morphologically and morphometrically similar to *H. stephanus*, except for spicule and gubernaculum lengths.

### 2.4. Hoplolaimus concaudajuvencus (Figure A2J, Figure A3J, Figure A4E,K and Figure A5, Table A4)

The topotype population of *H. concaudajuvencus* in our study was collected from St. Augustine grass and was compared morphologically with the population used for the original description by Golden and Minton [30] from Pensacola, Florida, parasitizing ryegrass (*Lolium multiflorum* Lam.) and Turfgreen bermudagrass.

Adult female. Females of the topotype population of this species had a cylindroid body shape like that reported in the original description by Golden and Minton [30]. They exhibited a lip region with four-seven lip annuli, a stylet longer than 50 µm with anteriorly dentate tulip-shaped knobs, and a pharyngeal gland lobe with three nuclei. Their hemizonid was located two annuli anterior to the excretory pore. Lateral field aerolated and marked by four incisures. Spermatheca conspicuous and filled with sperm. The vulva is like a transverse, deep slit. Pre-vulvar and post-vulvar phasmids are present. The tail is rounded as reported in the original description.

Adult male. The morphological characters of males were consistent with those in the original description.

Second-stage juvenile. The specimens of this life stage present in the topotype Florida population showed the same body characteristics described in the original description. They exhibited a conical pointed tail, unlike the rounded tails of the other juvenile life stages and adult females.

Remarks. Ellington and Golden [31] noted that this species exhibits morphological variations (heteromorphism) in the tail shape of females and juveniles; some adult specimens have a slight depression on the ventral side of the tail, and some have a terminal anal position. These authors also noted that the variation in *H. concaudajuvencus* juveniles consists of conically pointed tails, which change to the typical rounded hoplolaimid tails after the second molt. The presence of multiple phasmids was observed by these authors in 18 percent of the females and 15 percent of the males. As reported in the original description, the second-stage juveniles of the topotype population of this species showed a pointed tail terminus, unlike the rounded tail terminus of those in the species of *Hoplolaimus.* Females and males of our topotype population differed from those in the original description for having greater mean values of ratio c (69.5 and 48.8 vs. 65.1 and 37.9, respectively). The mean values of stylet and gubernaculum length of males of the topotype population were smaller than those in the original description (48.0 µm and 22.0 µm vs. 50.0 µm and 26.0 µm, respectively). Other characters of the topotype population did not differ from those reported in the original description, confirming the validity of the morphological identification of this topotype population as *H. concaudajuvenchus*.

### 2.5. Peltamigratus christiei (Figure A2L, Figure A3L and Figure A4F,L)

The Floridian population of this species obtained in this study was morphologically similar to those described by Golden and Taylor [32] and Sher [33].

### 2.6. Molecular Characterization, Phylogenetic Relationships, and Molecular Species Delimiting Hoplolaimus Species

Fifty-three new D2–D3 of 28S rRNA, 43 new ITS rRNA, and 47 new *COI* sequences from 23 isolates of *Hoplolaimus* spp. and one isolate of *Peltamigratus christiei* were obtained in this study.

#### 2.6.1. D2–D3 of the 28S rRNA Gene

The D2–D3 expansion segments of 28S rRNA gene sequence alignment contained 131 sequences of *Hoplolaimus*, one sequence of *P. christiei,* and two *Helicotylenchus* sequences included as outgroups. The alignment length was 699 bp. The phylogenetic relationships within sequences of *Hoplolaimus* species are given in Figure 3. There are three major clades in the phylogenetic tree: (i) *H. galeatus*, *H. magnistylus*, *H. floridensis* sp. n., *Hoplolaimus* sp. 1 (USA), and *Hoplolaimus* sp. B1 (previously identified as *H. concaudajuvencus*, USA); (ii) *H. concaudajuvencus;* and (iii) *H. seinhorsti*, *H. pararobustus*, *Hoplolaimus* sp. A (USA, California), and *H. tuberosus* Mwesige, Maosa, Couvreur & Bert 2025 (Uganda). Sequences of the topotype *Hoplolaimus galeatus* population clustered with sequences of nematodes previously identified by other authors as *H. stephanus.* All these ‘*H. stephanus’* sequences and other sequences of *Hoplolaimus* spp. clustered with sequences of the topotype *H. galeatus* population are considered here as representatives of *H. galeatus.* Sequences of the topotype *H. stephanus* population clustered with those of the topotype *H. galeatus.* Sequences of the lance nematodes from the Florida populations obtained in this study formed a clade with sequences of populations previously identified incorrectly as *H. galeatus*. All these sequences are considered representatives of a new species, *H. floridensis* sp. n. Sequences of the topotype *H. concaudajuvencus* population formed a separate clade within *Hoplolaimus.* Sequences of nematodes previously identified by some authors as *H. concaudajuvencus* are considered here as representatives of *Hoplolaimus* sp. B1. Sequences of *H. seinhorsti* from many countries around the world, including a new sequence from India obtained in this study, clustered with those of populations identified as *H. columbus* in South Carolina and southeastern states of the United States and those of populations identified as *H. indicus* from Iran, Morocco, and Myanmar. Sequences of populations of *H. magnistylus* from Ohio and Arkansas were distributed between two separate subclades. Sequences of *H. pararobustus* from Nigeria grouped in a separate clade. Maximal intraspecific sequence variations were for *H. galeatus*—1.6%, *H. magnistylus*—1.9%, *H. floridensis* sp. n.—0.2%, *H. concaudajuvencus*—0.4%, *H. seinhorsti*—0.7%, and *H. pararobustus*—2.9%.

#### 2.6.2. ITS of the rRNA Gene

The ITS rRNA gene sequence alignment contained 226 sequences of *Hoplolaimus*, two sequences of *P. christiei,* and two *Helicotylenchus* sequences included as outgroups. The alignment length was 948 bp. The phylogenetic relationships within *Hoplolaimus* species are given in Figure 4. There are several major clades in the phylogenetic tree: (i) *H. galeatus*, *H. magnistylus*, and *H. smokyensis*; (ii) *H. foridensis* sp. n.; (iii) *Hoplolaimus* sp. B1 (previously identified as *H. concaudajuvencus*, USA); (iv) *Hoplolaimus* sp. A (California); (v) *H. seinhorsti* and *H. tuberosus* (Uganda); and (vi) *H. concaudajuvencus*. Sequences of the topotype *Hoplolaimus galeatus* population clustered with sequences of nematodes previously identified by other authors as *H. stephanus.* All these ‘*H. stephanus’* sequences and other sequences of *Hoplolaimus* spp. clustered with sequences of the topotype *H. galeatus* population are considered here as representatives of *H. galeatus.* Sequences of the topotype *H. stephanus* population clustered with those of the topotype *H. galeatus.* Sequences of *H. seinhorsti* from many countries around the world, including a new sequence from India obtained in this study, clustered with those of populations identified as *H. columbus* in South Carolina and southeastern states of the United States, India, and Japan. Sequences of populations of *H. magnistylus* and *H. smokyensis* from temperate states in the United States clustered in well-separated clades, like those of a population of *P. christiei* from Florida. Maximal intraspecific sequence variations were for *H. galeatus*—15.7%, *H. magnistylus*—1.8%, *H. floridensis* sp. n.—1.4% (excluding PQ038855 and PQ038856) or 8.5% (with these two sequences), *H. concaudajuvencus*—1.2%, and *H. seinhorsti*—6.6%.

#### 2.6.3. COI Gene

The *COI* gene sequence alignment contained 224 sequences of *Hoplolaimus* and two *P. christiei* sequences included as outgroups. The alignment length was 308 bp. The phylogenetic relationships within *Hoplolaimus* species are given in Figure 5. There are several major clades in the phylogenetic tree: (i) *H. seinhorsti*, *H. tuberosus* (Uganda), *H. smokyensis*, (ii) *H. magnistylus*, (iii) *H. galeatus*, (iv) *H. foridensis* sp. n., *H. concaudajuvencus*, *Hoplolaimus* sp. B2 (previously identified as *H. concaudajuvencus*, USA), (v) *Hoplolaimus* sp. A (California), and (vi) *H. pararobustus*. Phylogenetic relationships within *Hoplolaimus* species were not well resolved. Sequences of the topotype *H. galeatus* population clustered with sequences of nematodes previously identified by other authors as *H. stephanus.* All these ‘*H. stephanus’* sequences are considered here as representatives of *H. galeatus.* Sequences of the topotype *H. stephanus* population clustered with those of the topotype *H. galeatus.* Sequences of the topotype *H. concaudajuvencus* population formed a separate clade within *Hoplolaimus.* Sequences of nematodes previously identified by some authors as *H. concaudajuvencus* from Texas (USA) are considered here as representatives of *Hoplolaimus* sp. B2. Sequences of *H. seinhorsti* from India and Israel clustered with those of *H. columbus* populations from the United States (Georgia and South Carolina) and *H. indicus* (Iran). Other sequences identified as *H. seinhorsti* from Indonesia and Pakistan formed a separate subclade like sequences of *H. tuberosus* and *H. smokyensis*. Sequences of *H. magnistylus* from Arkansas, Illinois, and Tennessee clustered in three separate subclades, like those of populations of *H. pararobustus* from different localities in Nigeria. The sequences of a population of *P. christiei* from Florida clustered in a separate clade. Maximal intraspecific sequence variations were for *H. galeatus*—12.9%, *H. magnistylus*—10.4%, *H. floridensis* sp. n.—6.5%, *H. concaudajuvencus*—0.3%, *H. seinhorsti*—13.1% (0.6% for North America), and *H. pararobustus*—7.5%.

#### 2.6.4. Combined rRNA and COI Gene Alignment

The combined rRNA and *COI* gene sequence alignment contained reference sequences of 8 valid and 4 unidentified *Hoplolaimus* species, one *P. christiei* sequence, and two sequences of two *Helicotylenchus* species used as outgroup taxa. The alignment length was 1891 bp. The phylogenetic relationships within *Hoplolaimus* species are given in Figure 6. *Hoplolaimus concaudajuvencus* occupied a basal position, and all other *Hoplolaimus* sequences were distributed into two major clades: (i) *Hoplolaimus* sp. A—*Hoplolaimus* sp. 1 and (ii) *H. parorobustus*—*H. seinhorsti* (Figure 6).

### 2.7. Distribution Map of Hoplolaimus Species in the United States

Distribution maps of eight *Hoplolaimus* species are given in Figure 7. Maps represent a large portion of the United States delimited by the Great Lakes to the north, the Atlantic Ocean to the east, the Gulf of Mexico to the south, and the Mississippi River System to the west. Only reports confirmed by molecular identification were included and marked as colored rounded dots. Putative areas of distribution of the species are delimited by puncture or dash lines.

## 3. Discussion

### 3.1. Identification of Hoplolaimus galeatus and H. stephanus syn. n.

The type species, *H. galeatus,* was described from Arlington Farms, Arlington, Virginia. A history of the description of this species has been provided by Sher [29,34]. Together with the redescription of *H. galeatus*, Sher [29] also described three new species, *H. stephanus*, *H. columbus,* and *H. indicus.*

Mullin et al. [35] were the first to deposit sequences of a short fragment of the 18S rRNA gene for a population of *H. galeatus* collected in Konza, Kansas, in the GenBank. These sequences were not mentioned in their paper that deals with non-plant-parasitic nematodes. The lance nematodes identified as *H. galeatus* from several states were further characterized using the ITS1 rRNA and the D1–D3 expansion segments of 28S rRNA gene sequences by Bae et al. [16], the ITS1 rRNA by Ma et al. [18], the 18S rRNA by Zheng et al. [36], and the ITS rRNA and *COI* by Holguin et al. [20]. The D2–D3 of 28S, ITS rRNA, and *COI* gene sequences of the topotype *H. galeatus* population obtained in this study did not cluster with sequences of *H. galeatus* published by the above-mentioned authors but formed a clade with the lance nematodes originally identified as *H. stephanus* by Ma et al. [18] and Holguin et al. [20]. Thus, we consider all ‘*H. stephanus’* sequences clustered with the sequences of the topotype *H. galeatus* population as representatives of *H. galeatus.*

*Hoplolaimus stephanus* was described from swamp soils in Nichols, South Carolina, and Sher [29] distinguished it from *H. galeatus* by smaller numbers of longitudinal striations on the basal annulus (24.0–28.0 vs. 32.0–36.0) of the lip region, shorter gubernaculum (13.0–20.0 vs. 20.0–28.0 µm) and spicules (30.0–38.0 vs. 40.0–52.0 µm), usually less areolation of the lateral field, and smaller body size. This species has never been redescribed, and there is no information on its morphological and morphometrical variations, except for the study by Vovlas et al. [37]. These authors did not find topotype specimens of *H. stephanus* and described a population of putative *H. stephanus* from Raleigh, North Carolina (USA). This population had 4 lip annuli like those reported for this species. However, other differential characters overlapped with those of topotype *H. galeatus* in our study. These characters included striations on the basal annulus (30.0–36.0 vs. 32.0–36.0), body length (1008.0–1429.0 vs. 1038.0–1300.0 µm), spicule length (35.0–42.0 vs. 42.0–45.0 µm), and gubernaculum length (16.0–21.0 vs. 21.0–23.0 µm). The discrepancies in these diagnostic characters reported by Vovlas et al. [37] and those by Sher [29] indicate that this population from North Carolina may be *H. galeatus*.

The results of molecular analyses showed that the D2–D3 of 28S rRNA, ITS rRNA, and *COI* gene sequences of the nematodes collected from the type locality of *H. stephanus* are similar to those of the topotype population of *H. galeatus*. Taking into account intraspecific variability of longitudinal striation numbers on the basal annulus and areolation in the lateral field and small differences in gubernaculum and spicule lengths and morphological similarity of these two species, *H. stephanus* syn. n. is considered a synonym of *H. galeatus.*

### 3.2. Identification of Hoplolaimus floridensis sp. n. and H. tylenchiformis

Sher [29] indicated that Whitten [38] proposed a new species, *Hoplolaimus neocoronatus,* from Florida in a thesis and considered it as a *nomen nudum*. Sher [29] also concluded that although specimens of this species were unavailable for study, they appeared to be similar to *H. tylenchiformis* on the basis of the description. There are several reports of the presence of *H. tylenchiformis* in the United States, including Florida [39,40,41,42,43,44]. However, all these records were considered unconfirmed or later referred to *H. galeatus* findings.

Since Floridian local and non-local samples initially identified as *H. galeatus* are morphologically and molecularly different from *H. stephanus* and, at the same time, do not match those of topotype *H. galeatus*, they represent another *Hoplolaimus* species, which is described here as *H. floridensis* sp. n.

### 3.3. Identification of Hoplolaimus concaudajuvencus

*Hoplolaimus concaudajuvencus* was described by Golden and Minton [30] from soil and roots of golf green turf, consisting of annual ryegrass (*Lolium multiflorum*) and “Tifgreen” Bermuda grass (*Cynodon dactylon*) on the Pensacola Country Club Golf Course, Pensacola, Florida. This species was considered to be different from the most closely related *H. galeatus* by the presence of a conically pointed tail in first- and second-stage juveniles, a longer stylet with knobs that appeared to be more accentuated in a tulip shape, a markedly dentate anterior that was more appressed to the stylet shaft than in *H. galeatus,* and the presence of multiple phasmids [31]. Bae et al. [16] provided the sequences of the ITS1 and D1–D3 of 28S RNA genes for lance nematodes identified as *H. concaudajuvencus* from Perry County, Arkansas. Later, Holguin et al. [20] also published sequences of the ITS rRNA and *COI* genes for several samples identified as *H. concaudajuvencus* from Dallas County, Texas. Sequences of the Arkansas population were different from all other sequences and from that of the topotype *H. concaudajuvencus* population obtained in our study and assigned here to a new species indicated as *Hoplolaimus* sp. B1. Sequences of *COI* of the Texas population were also different from all other sequences and from that of the topotype *H. concaudajuvencus* population; however, the ITS sequences clustered with those of *H. magnistylus* due to possible contamination. Because we could not compare these two populations molecularly, the Texas samples were assigned to another new species indicated as *Hoplolaimus* sp. B2.

### 3.4. Identification of Hoplolaimus seinhorsti

*Hoplolaimus seinhorsti* was described by Luc [45] from rhizosphere soil and roots of cotton plants growing in Miary, near Tulear, Madagascar, and morphologically characterized in more detail by several authors [4,29,46,47]. Later, Sher [29] described *H. columbus* from soil rhizosphere soil of soybeans in Eastover, South Carolina, USA. He noticed that *H. columbus* was different from the closely related species *H. seinhorsti* by the lower position of the excretory pore and the conspicuous overlap of the intestine over the rectum. In the same article, Sher also described another species, *H. indicus,* from rhizosphere soil of sugarcane in Karnal, Punjab, India. *Hoplolaimus indicus* was distinguished from the closely related species *H. columbus* by the shorter female tail, the lower position of the excretory pore, smaller size, shorter spear, and the presence of a spermatheca in the females and males, and in the face view of the basal annulus of the head region, having 6 to 12 longitudinal striations. *Hoplolaimus columbus* and *H. indicus* were identified in many world regions and morphologically characterized by several authors [14,23,24,48,49,50,51,52]. The distribution and host-plant range of these three *Hoplolaimus* species are overlapping [53]. The results of our present molecular analysis show a high sequence similarity between the three species in three gene regions that agreed with the results of phylogenetic and sequence analyses from other publications [4,23,24,51] and indicate that sequence variation between these three species is in a range of intraspecific variation for this group. Taking into consideration the high morphological and morphometrical variations of these *Hoplolaimus* species and their molecular similarity, we propose to synonymize *H. columbus* syn. n and *H. indicus* syn. n with *H. seinhorsti.* Siddiqi [13] noticed that further studies of a large number of *Hoplolaimus* species described in South Asia are needed due to their similarity with *H. seinhorsti.*

### 3.5. Unidentified Hoplolaimus Species

Present molecular analysis revealed the presence of several *Hoplolaimus* sequences, which belong to unidentified species of this genus. *Hoplolaimus* sp. A was found from a sample collected under trees in California, Riverside County, and likely belongs to *H. californicus*, which is the only species of this genus described and found in several locations in Southern California [29]. The small number of specimens of *Hoplolaimus* sp. A. in the sample was insufficient for both morphological and molecular analyses to verify this assumption, since no DNA sequences are available for *H. californicus*. Sequences of *Hoplolaimus* sp. B1 and *Hoplolaimus* sp. B2 were originally identified as those of *H. concaudajuvencus* by Bae et al. [17] from Arkansas and Holguin et al. [20] from Texas, respectively. Co-specificity of these two samples should be confirmed by further morphological and molecular study. *Hoplolaimus* sp. 1, found in Smoky Mountain, TN, USA, was only molecularly characterized by Bae et al. [17].

### 3.6. Subgenera of Hoplolaimus

Phylogenetic analysis of the D2–D3 of the 28S rRNA gene sequences provided some resolution between species compared with less resolved the ITS rRNA and *COI* gene phylogenies. However, this phylogeny did not reveal groupings of different *Hoplolaimus* species under the genus level as it has been proposed by Siddiqi [13]. Thus, the molecular results of this study do not justify the division of this genus into subgenera.

### 3.7. Biogeography of Hoplolaimus Species in the United States

Several *Hoplolaimus* species showed high intraspecific variations in *COI* gene sequences; however, they were in variation ranges for some tylenchid nematodes found in the centers of diversification. For example, maximal intraspecific *COI* gene sequence diversity for *G. pallida* and *G. rostochiensis* was estimated at 20.7% and 14.2%, respectively, in the Andes, Peru, and Bolivia [54]; for *G. mexicana,* it was 9.9% in the Sierra Madre mountain systems, Mexico [55]; and for *H. filipievi,* it was 10.6% in the Irano-Anatolian region, Iran [56], which are the centers of origins and diversification of these species. Based on variations in *COI* gene sequences for *Hoplolaimus*, we can suggest several centers of origins and diversification for this genus: (i) Eastern regions of North America for *H. concaudajuvencus*, *H. floridensis* sp. n., *H. galeatus*, *H. magnistylus*, and some other *Hoplolaimus* spp.; (ii) South and Southeast Asia for *H. seinhorsti*; (iii) Africa for *H. pararobustus;* and (iv) Southwestern regions of North America (California and Mexico) for *Hoplolaimus* sp. A., *H. californicus*, *H. igualaensis,* and *H. maggentii* Cid del Prado 1994.

Phylogeographical patterns for *Hoplolaimus* species in the Southern region of North America represent several geographically structured and genetically divergent lineages. The distribution of these lineages well matches the patterns found for many studies of plants and animals in North America [57].

The Appalachian Mountains in the Southeastern United States could be suggested as a center of origins and diversification of *H. galeatus*. The Appalachian Mountains stretch from Alabama in the south to New York in the north and are considered the oldest mountains on Earth, with the richest biodiversity among temperate areas. The mountains have a wide variety of soils, climates, and topographies, imposing barriers that isolate populations. During the Pleistocene Epoch, the *H. galeatus* populations could have been located in several mountain refugia and then expanded to other regions after deglaciation.

*Hoplolaimus galeatus* has a complex and differentiated *COI* haplotype structure, with the basal clade containing populations from South Carolina, Maryland, and Pennsylvania. From the Appalachians and the coastal plain of the Carolinas, this species likely dispersed in northeast, northwest, and west directions. *Hoplolaimus galeatus* was reported with molecular confirmation in North Carolina, Georgia, Missouri, Indiana, Massachusetts, Iowa, North Dakota, Kansas, Nebraska, and Ohio. It is the most widely distributed species in regions from subtropical to continental climates with distinct seasons, with maximal northern distribution in the United States. There is one report of *H. galeatus* from Canada in Brandon, Manitoba. The report of the lance nematode identified by Anderson [58] in Canada as *H. indicus* likely also belongs to *H. galeatus*. Identifications of *H. galeatus* outside North America in Australia, Central and South America, Asia, and Africa [53,59,60] require confirmation.

*Hoplolaimus floridensis* sp. n. showed a high *COI* haplotype diversity in peninsular Florida, where this species genetically diversified and dispersed to South Carolina and Alabama. Southern refugia for *H. magnistylus* were likely located in the Gulf Coast region, from which this species dispersed north along the Mississippi Valley. The Florida Panhandle is the only region where *H. concaudajuvencus* has been reported.

Populations of *H. seinhorsti* showed minimal *COI* gene sequence differences in the United States, compared with those in South and Southeast Asia. It could be proposed that this species originated from Asian regions and then relatively recently dispersed across the world and was introduced in North America, where it is presently reported in Georgia, North Carolina, South Carolina, Louisiana, and northern Florida. Wide distribution in tropical and subtropical world regions could be explained by the possibility of parthenogenetic reproduction for this species; absence or small male numbers have been observed in some populations of this species. This type of reproduction mode significantly increases chances of successful colonization in new areas.

The southwestern region of North America, which includes California and Mexico, could be another center of *Hoplolaimus* diversification. Presently several valid species, *H. californicus*, *H. maggentii,* and *H. igualaensis,* were described morphologically and without molecular data from this region. In our study, *Hoplolaimus* sp. A from California represents a unique lineage within *Hoplolaimus*. Molecular studies of these species would further elucidate the phylogenetic relationships among these species and their diversification patterns. The lack of molecular data on *H. tylenchiformis* prevented any biogeographic analysis of this elusive species.

### 3.8. Hoplolaimus Species from Florida

No less than five species of *Hoplolaimus* have been reported in Florida [44], but their identification has not been confirmed by subsequent morphological and molecular analyses. These species included *H. concaudajuvencus*, *H. galeatus*, *H. magnistylus*, *H. seinhorsti,* and *H. tylenchiformis.* No Florida populations of *H. tylenchiformis*, *H. magnistylus,* and *H. seinhorsti* were available for morphological analyses for this study. The reports of the presence of *H. tylenchiformis* in Florida include a collection of a population around seedlings of *Citrus* sp. by D.J. Raski and A.C. Tarjan in an undetermined location [42] and an unconfirmed detection in Alachua County [44]. This author also made an unconfirmed report of *H. magnistylus* in the same county. Information about the presence of *H. seinhorsti* in Florida was provided by R.A. Kinlock, who observed a decline in the populations of this species under a crop system that included soybean. Occurrence of *H. seinhorsti* in cornfields and pastures in Escambia County was also reported by Van den Berg [46]. Bae et al. [16] molecularly confirmed the presence of *H. seinhorsti* from peanut in the IFAS Experimental Station, Jay County. In this study, we have not found populations of *H. galeatus* from Florida, where *H. floridensis* sp. n. is the most common lance nematode occurring in the state.

Considering the new molecular identity for *H. seinhorsti* and *H. galeatus*, its junior synonym, *H. stephanus* syn. n., and the possible misidentification of the *Hoplolaimus* species in the literature, careful study is needed to verify published information on the biology, ecology, pathogenicity, and PCR diagnostic methods of the species of this genus.

## 4. Materials and Methods

### 4.1. Nematode Samples and Extraction

The nematode populations used in this study were obtained from soil and root samples, each consisting of 25 plugs of 20 cm^3^. The topotype population of *H. galeatus* was collected from Arlington National Cemetery (latitude—38°52′28.4″ N, longitude—77°03′49.28″ W), Virginia, in fall of 2021; the topotype *H. concaudajuvencus* population was extracted from soil samples from Pensacola Country Club, Pensacola (latitude—30°38′59.5″ N, longitude—87°26′2.02″ W), Florida, in 2022; and the topotype *H. sephanus* population was collected from Nichols (latitude—34°13′22.8″N, longitude—79°07′48.0″), South Carolina, in spring of 2025. The sugar centrifugal flotation method was used to extract nematodes from soil [61]. The list of studied species and populations is given in Table 1.

### 4.2. Morphological Study and Species Identification

Nematode specimens were killed and fixed in 3% formaldehyde and processed to glycerin using the formalin-glycerin method [62,63]. The specimen measurements were taken using an ocular micrometer on a Leica WILD MPS48 attached to a Leitz DMRB compound microscope. All measurements are in micrometers unless specified. Light micrographs of nematodes were taken with a Zeiss Axio Imager 2 (Carl Zeiss Microscopy, LLC, New York, NY, USA) using a Zeiss Axio Cam 712 color camera (Carl Zeiss Microscopy, LLC, New York, NY, USA). Images were processed using the Zeiss ZEN 3.6 blue edition software.

For scanning electron microscopy (SEM), nematodes were fixed using the sequential fixation according to Eisenback [64]. Briefly, nematode specimens were transferred to a 1.5 mL Eppendorf tube containing 0.5 mL of distilled water and kept at 4 °C for two hours. Specimens were subsequently fixed by adding four drops of fixative solution (4% glutaraldehyde and 2% formalin buffered with 0.1 M sodium cacodylate pH 7.2) every half hour until the fixative reached a final concentration of 2% glutaraldehyde and 1% formalin, followed by an overnight fixation at 4 °C. Specimens were stained in 2% osmium tetroxide for one hour, followed by a triple 5 min rinse in 0.1 M sodium cacodylate pH 7.2 before they were transferred to a 13 mm Swinnex^®^ filter holder (MilliporeSigma, Burlington, MA, USA), with a 0.2 μm microfiber prefilter disc, and dehydrated through an ethanol series (25%, 50%, 70%, 85%, 95%, and 2× 100%) for 5 min each. The microfiber filter holder containing the specimens was transferred to CPD carriers and placed in a glass jar filled with 100% ethanol. Critical Point Drying (CPD) was performed using a Leica EM CPD300 (Leica Microsystems, Deerfield, IL, USA) according to the manufacturer specifications. Samples were mounted to 12 mm aluminum mount stubs covered with conductive carbon PELCO Tabs (Ted Pella Inc., Redding, CA, USA), and sputter coated with 5 nm of iridium using a Leica EM ACE 600 Sputter Coater. Images were captured with a Zeiss Sigma Field Emission Scanning Electron Microscope (Zeiss GmBh, New York, NY, USA).

Species delimitation of the lance nematode populations in this study was performed using an integrated approach based on morphological and morphometric evaluation combined with molecular-based phylogenetic inference (tree-based methods) and sequence analyses (genetic distance methods).

### 4.3. DNA Extraction, PCR, and Sequencing

DNA was extracted from several specimens of each sample using the proteinase K protocol. DNA extraction, PCR, and cloning protocols were used as described by Subbotin [65]. The following primer set was used for PCR: forward D2A (5′-ACA AGT ACC GTG AGG GAA AGT TG-3′) and reverse D3B (5′-TCG GAA GGA ACC AGC TAC TA-3′) primers for amplification of the D2–D3 expansion segments of the 28S rRNA gene; forward TW81 (5′-GTT TCC GTA GGT GAA CCT GC-3′) and reverse AB28 (5′-ATA TGC TTA AGT TCA GCG GGT-3′) primers for amplification of the ITS1-5.8-ITS2 rRNA gene; and forward JB3 (5′-TTT TTT GGG CAT CCT GAG GTT TAT-3′) and reverse JB4 (5′-TAA AGA AAG AAC ATA ATG AAA A-3′) or JB5 (5′-AGC ACC TAA ACT TAA AAC ATA ATG AAA ATG-3′) primers for amplification of the partial *COI* gene of mtDNA. The PCR products were purified using QIAquick (Qiagen, Germantown MD, USA) Gel and submitted for direct sequencing, or cloned using the pGEM-T Vector System II kit (Promega, Madison, WI, USA), and PCR from clones was also purified and submitted for sequencing. The sequencing was conducted at Azenta (Chelmsford, MA, USA). The newly obtained sequences were submitted to the GenBank database under accession numbers: PV863224-PV863266 (ITS rRNA gene), PV863155-PV863206 (D2–D3 of the 28S rRNA gene), and PV857615-PV857661 (*COI* gene), as indicated in Table 1 and the phylogenetic trees.

### 4.4. Phylogenetic and Sequence Analysis

The newly obtained sequences of the D2–D3 expansion segments of the 28S rRNA, the ITS of the rRNA, and partial *COI* mtDNA genes were aligned using ClustalX 1.83 [66] with corresponding published gene sequences [4,5,14,15,16,17,18,19,20,21,22,23,24,25,51,67,68,69]. Alignments for the D2–D3 of 28S rRNA, the *COI* gene, and the ITS rRNA gene sequences were generated with default parameters. Outgroup taxa for each dataset were chosen based on previously published data [15]. Alignments were manually edited using GenDoc 2.5 [70], and a few ambiguous nucleotides in conserved regions of several sequences were removed, or the sequences with a large number of such nucleotides were deleted from the analysis. Alignments were analyzed with Bayesian inference (BI) using MrBayes 3.1.2 [71]. BI analysis for each gene was initiated with a random starting tree and was run with four Metropolis-coupled Markov chain Monte Carlo (MCMC) analyses for 1.0 × 10^6^ generations. The MCMC analyses were sampled at intervals of 100 generations. Two runs were performed for each analysis. After discarding burn-in samples and evaluating convergence, the remaining samples were retained for further analyses. The topologies were used to generate a 50% majority rule consensus tree as described by Subbotin [72]. Posterior probabilities (PPs) are given on appropriate clades. The best-fit models of DNA evolution were obtained using the program jModelTest 0.1.1 [73] with the Akaike Information Criterion. Pairwise divergence between sequences was calculated using PAUP*4a [74].

### 4.5. Distribution Map of Hoplolaimus Species in the United States

Original and published information was used to reconstruct distribution maps for *Hoplolaimus* species in the United States. Species selected for this biogeographical study were all identified by molecular methods in this study and in the literature [16,17,18,19,20,22,25,67].

## Figures and Tables

**Figure 1 ijms-26-08501-f001:**
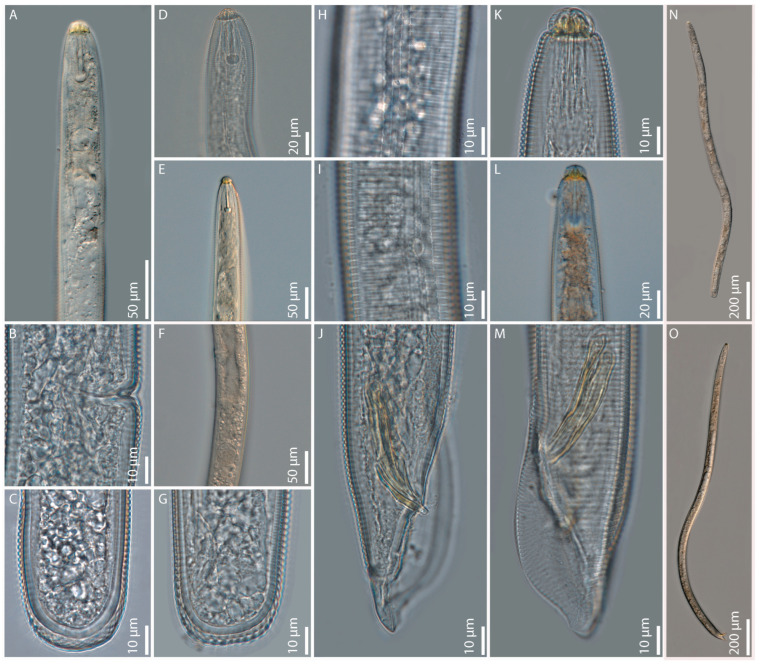
Light microphotographs of *Hoplolaimus floridensis* sp. n.: (**A**) female pharyngeal region; (**B**) female vulvar region; (**C**,**G**) female posterior ends; (**D**,**E**) female anterior region; (**F**) female vulvar region with spermatheca; (**H**,**I**) female lateral field with phasmid; (**J**,**M**) male posterior region showing spicules, gubernaculum, and bursa; (**K**,**L**) Male anterior ends; (**N**) female entire body; and (**O**) male entire body.

**Figure 2 ijms-26-08501-f002:**
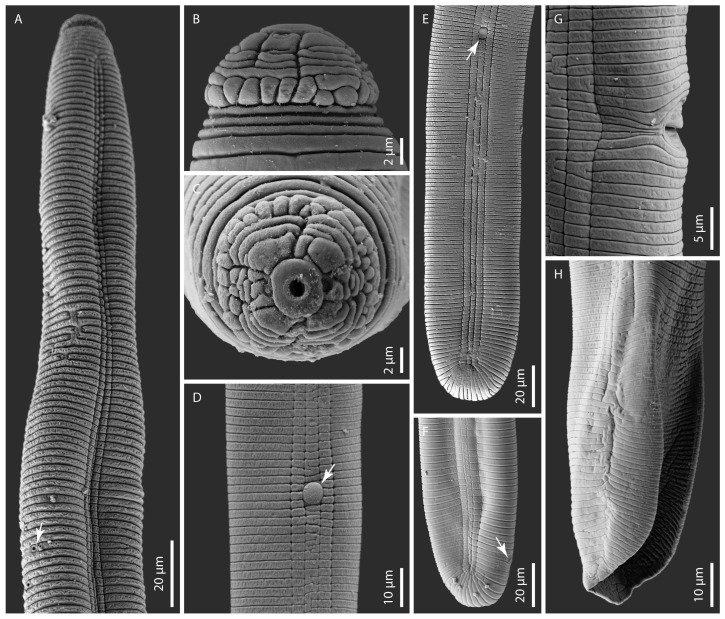
Scanning electron microphotographs of *Hoplolaimus floridensis* sp. n. (**A**) female anterior region showing annuli, lateral field, and excretory pore (arrowed) and (**B**,**C**) female head patterns in lateral and *en face* views. Note the basal lip annulus divided into superimposed blocks and incomplete lip annuli with various subdivisions in (**B**). Note a prominent and rounded oral disc with a round stoma surrounded by divided sub-dorsal, sub-ventral, and fragmented lateral lip sectors, which delimit the small pocket-like amphidial apertures in (**C**); (**D**) lateral field with a prominent phasmid (arrowed); (**E**,**F**) female posterior region showing lateral field marked by four incisures. Note the phasmid (arrowed) in (**E**) and the anus (arrowed) in (**F**); (**G**) lateral view of the vulval region showing a portion of the slit-like vulva; and (**H**) the male posterior region showing the bursa alae.

**Figure 3 ijms-26-08501-f003:**
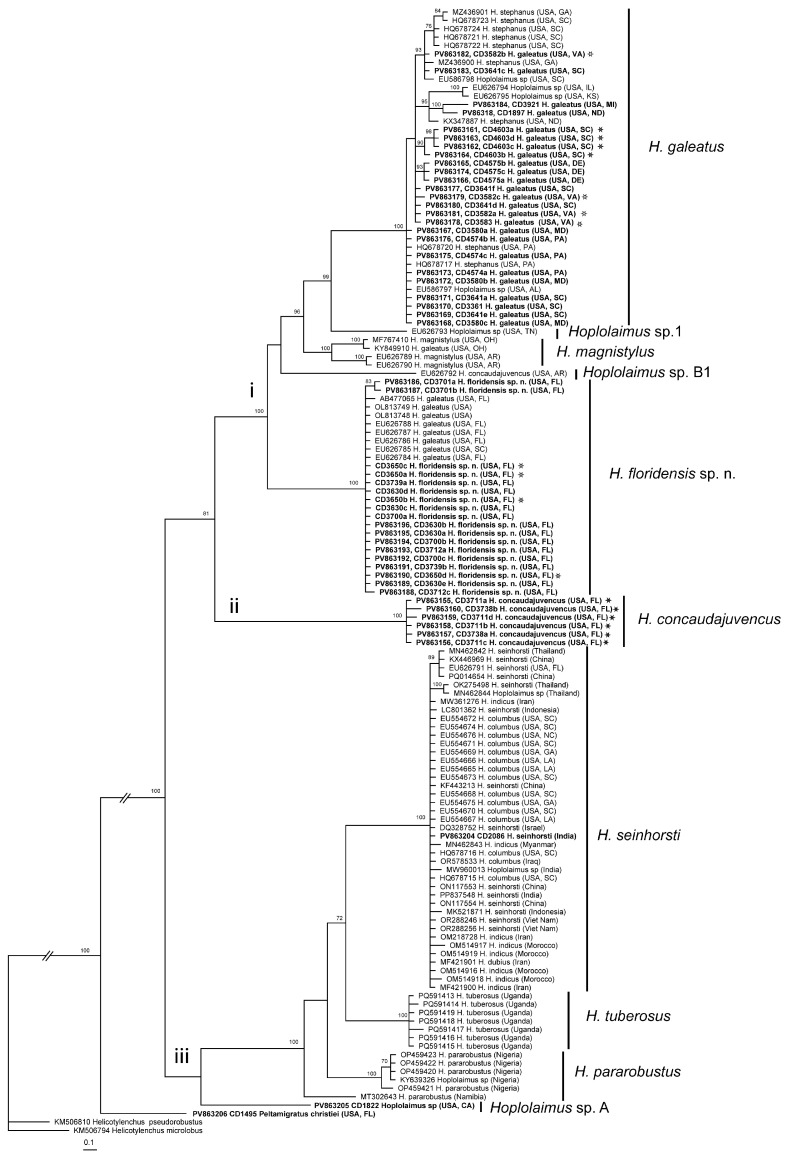
Phylogenetic relationships within *Hoplolaimus* species as inferred from Bayesian analysis of the D2–D3 of the 28S rRNA gene sequences under the GTR + I + G model. Posterior probabilities equal to or greater than 70% are given for appropriate clades. Original species identifications are given for all sequences. The new sequences are indicated in bold. Small-font Latin species names are original identifications. Species names in a large font with bars are species delimitations made in this study. Sequences from the topotype populations are indicated by star symbols. Major clades are indicated by lowercase Roman numerals.

**Figure 4 ijms-26-08501-f004:**
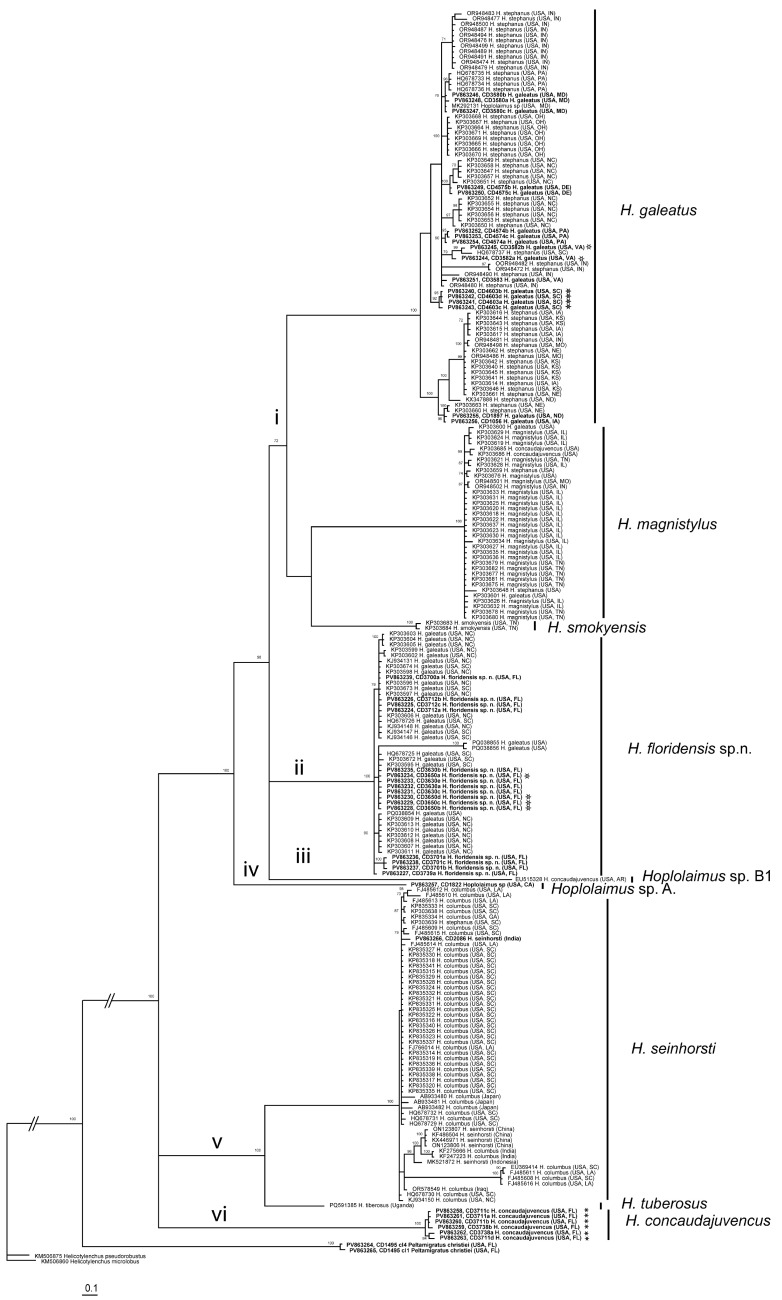
Phylogenetic relationships within *Hoplolaimus* species as inferred from Bayesian analysis of the ITS rRNA gene sequences under the GTR + I + G model. Posterior probabilities equal to or greater than 70% are given for appropriate clades. Original species identifier given for all sequences. The new sequences are indicated in bold. Small-font Latin species names are original identifications. Species names in a large font with bars are species delimitations made in this study. Sequences from the topotype populations are indicated by star symbols. Major clades are indicated by lowercase Roman numerals.

**Figure 5 ijms-26-08501-f005:**
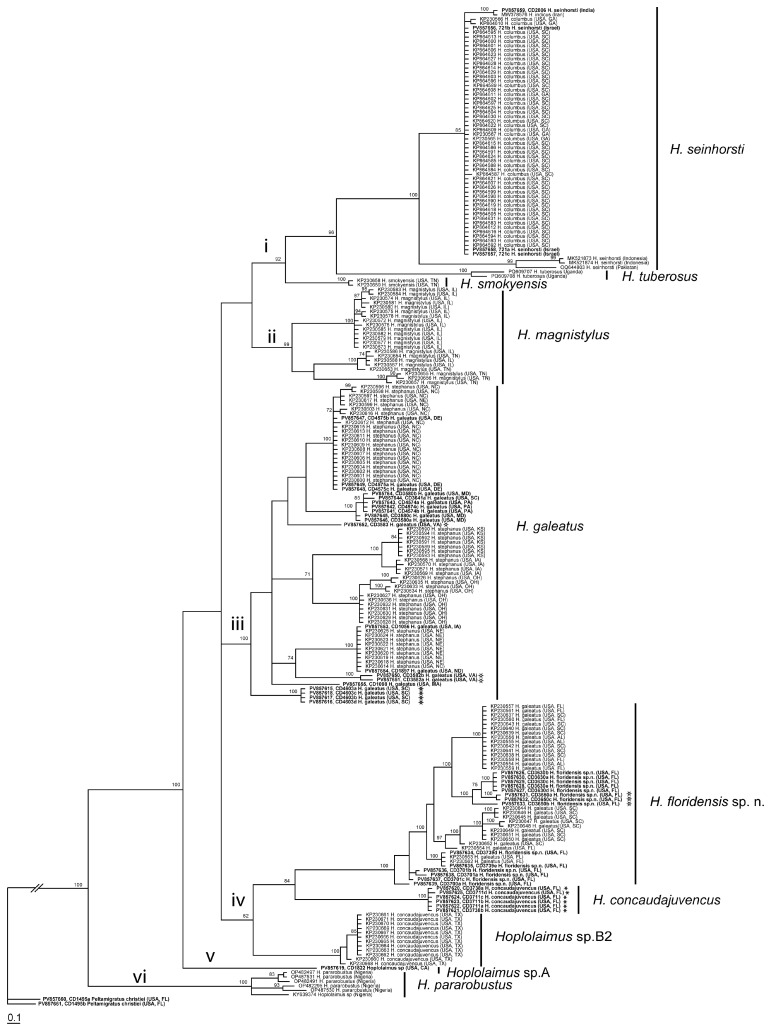
Phylogenetic relationships within *Hoplolaimus* species as inferred from Bayesian analysis of the *COI* gene sequences under the GTR + I + G model. Posterior probabilities equal to or greater than 70% are given for appropriate clades. Original species identifications are given for all sequences. The new sequences are indicated in bold. Small-font Latin species names are original identifications. Species names in a large font with bars are species delimitations made in this study. Sequences from the topotype populations are indicated by star symbols. Major clades are indicated by lowercase Roman numerals.

**Figure 6 ijms-26-08501-f006:**
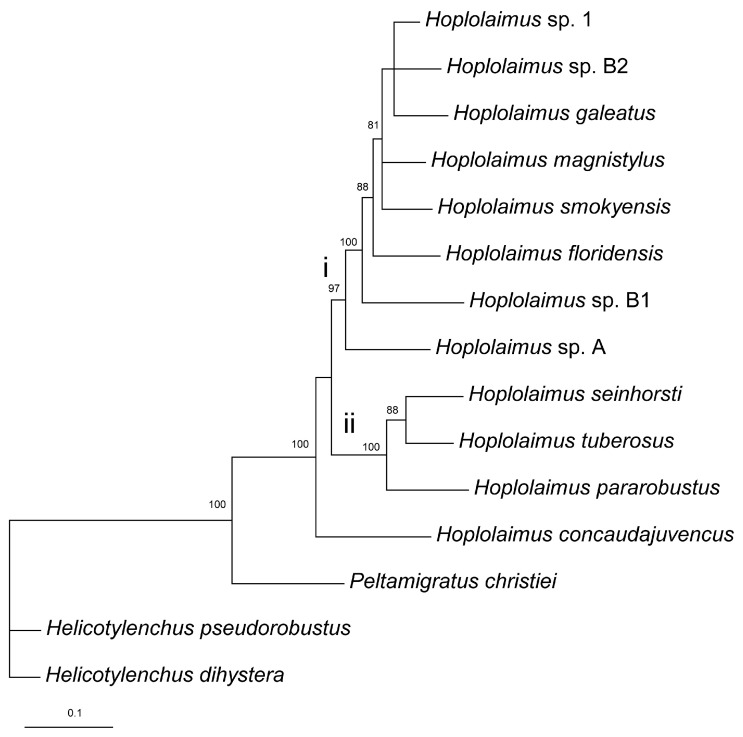
Phylogenetic relationships within *Hoplolaimus* species as inferred from Bayesian analysis of the D2–D3 of 28S rRNA, ITS rRNA, and *COI* gene sequences under the GTR + I + G model. Posterior probabilities greater than 70% are given for appropriate clades. Major clades are indicated by lowercase Roman numerals.

**Figure 7 ijms-26-08501-f007:**
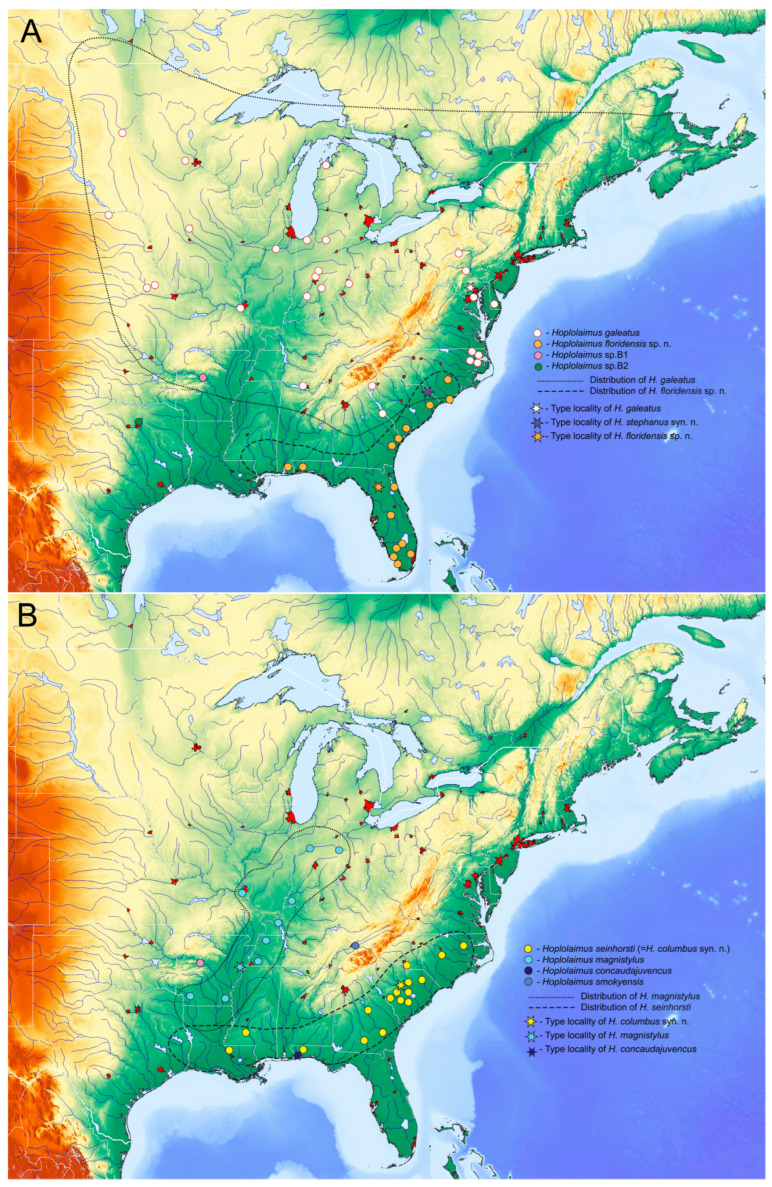
Distribution of several *Hoplolaimus* species in eastern regions of North America reconstructed based on published and original molecular identification of samples. (**A**) *H. galeatus*, *H. floridensis* sp. n., *Hoplolaimus* sp. B1, *Hoplolaimus* sp. B2; (**B**) *H. seinhorsti*, *H. magnistylus*, *H. concaudajuvencus*, *H. smokyensis*.

**Table 1 ijms-26-08501-t001:** *Hoplolaimus* species and populations characterized in this study.

Species	Location	Associated Plants	Sample Code	GenBank Accession Number			Source or Reference
				ITS rRNA Gene	D2–D3 of 28S rRNA Gene	*COI* Gene	
*H. concaudajuvencus*	USA, Florida, Escambia County, Pensacola	*Stenotaphrum secundatum*, *Lolium* sp.	CD3711; 02152022-01216	PV863258, PV863260, PV863261, PV863263	PV863155, PV863156, PV863158, PV863159	PV857622-PV857625	L. Violett, R.N. Inserra
*H. concaudajuvencus*	USA, Florida, Escambia County, Pensacola	*S. secundatum*	CD3738; 03292022-02584	PV863259, PV863262	PV863157, PV863160	PV857620, PV857621	L. Violett, R.N. Inserra
*H. galeatus*	USA, Virginia, Arlington cemetery	Grasses	CD3582	PV863244, PV863245	PV863179, PV863181, PV863182	PV857650, PV857651	M. Kantor
*H. galeatus*	USA, Virginia, Arlington cemetery	Grasses	CD3583	PV863251	PV863178	PV857652	M. Kantor
*H. galeatus* (*=H. stephahus* syn. n.)	USA, South Carolina, Marion County, Nicols	Unknown	CD4603; sample 5	PV863240-PV863243	PV863161-PV863164	PV857615-PV857618	M. Kantor
*H. galeatus*	USA, Michigan, Leelanau County	Hop	CD3921	-	PV863184	-	M. Quintanilla-Tornel
*H. galeatus*	USA, Maryland, Prince George’s County, Upper Marlboro	Grasses	CD3580	PV863246-PV863248	PV863167, PV863168, PV863172	PV857640, PV857645, PV857646	M. Kantor
*H. galeatus*	USA, Maryland, Prince George’s County, Upper Marlboro	Grasses	CD3361	-	PV863170	-	M. Kantor
*H. galeatus*	USA, Iowa	Corn	CD1056	PV863256	-	PV857653	G. Tylka
*H. galeatus*	USA, South Carolina	Unknown	CD3641	-	PV863169, PV863171, PV863177, PV863180, PV863183	PV857644	Z. Handoo
*H. galeatus*	USA, North Dakota	Pea	CD1897	PV863255	PV863185	PV857654	G.P. Yan
*H. galeatus*	USA, Minnesota, Wright County, Monticello	Grasses	CD1098	-	-	PV857655	D. Mollov
*H. galeatus*	USA, Pennsylvania, Centre County, Rock Springs	Corn	CD4574	3PV863252-PV863254	PV863173, PV863175, PV863176	PV857641-PV857643	M. Kantor
*H. galeatus*	USA, Delaware, Sussex County, Ellendale	Corn	CD4575	PV863249, PV863250	PV863165, PV863166, PV863174	PV857647-PV857649	M. Kantor
*H. seinhorsti*	Israel	Unknown	721	-	DQ328752	PV857656-PV857658	N. Vovlas; Subbotin et al. [15]
*H. seinhorsti*	India, Uttar Pradesh	Grasses	CD2086	PV863266	PV863204	PV857659	S.A. Subbotin
*H. floridensis* sp. n.	USA, Florida, Alachua County, Gainesville	*S. secundatum*	CD3630; 10222021-3125	PV863231-PV863233, PV863235	PV863189, PV863195, PV863196, PV863198, PV863200	PV857626-PV857630	R.N. Inserra
*H. floridensis* sp. n.	USA, Florida, Marion County, Citra	*Cynodon dactylon*	CD3650; 11152021-04429	PV863228-PV863230, PV863234	PV863190, PV863199, PV863202, PV863203	PV857631-PV857633	W.W. Crow, R.N. Inserra
*H. floridensis* sp. n.	USA, Florida, Collier County, Chokoloskee	Grasses	CD3700; 12202021-03392	PV863239	PV863192, PV863194, PV863197	PV857639	J.S. Stanley, R.N. Inserra
*H. floridensis* sp. n.	USA, Florida, Collier County, Everglades City	Grasses	CD3701; 12202021-03390	PV863236-PV863238	PV863186, PV863187	PV857636-PV857638	J.S. Stanley, R.N. Inserra
*H. floridensis* sp. n.	USA, Florida, Hendry County, Clewiston	*Zoysia* sp. *+ S. secundatum*	CD3712; 02172022-01316	PV863224-PV863226	PV863188, PV863193	-	J.S. Stanley, R.N. Inserra
*H. floridensis* sp. n.	USA, Florida, Collier County, Immokalee	*S. secundatum*	CD3739; 05202022-04589	PV863227	PV863191, PV863201	PV857634, PV857635	J. S. Stanley, R.N. Inserra
*Hoplolaimus* sp. A	USA, California, Riverside County, Temecula, Spring Canyon	Unknown	CD1822	PV863257	PV863205	PV857619	S.A. Subbotin
*Peltamigratus christiei*	USA, Florida, Okeechobee County, Okeechobee	Palm	CD1495; 06232022-05749	PV863264, PV863265	PV863206	PV857660, PV857661	R. Bloom; R.N. Inserra

## Data Availability

Sequencing data are available from the NCBI database, accessions # PV863224-PV863266 (ITS rRNA gene), PV863155-PV863206 (D2–D3 of the 28S rRNA gene), and PV857615-PV857661 (*COI* gene). All other relevant data are included within the manuscript.

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
