# Peer review of "Molecular Phylogeny and Species Delimiting for the Genus Hoplolaimus (Nematoda: Tylenchida) with Description of Hoplolaimus floridensis sp. n. and Notes on Biogeography of the Genus in the United States"

_ijms, 2025, doi:10.3390/ijms26178501_

Round 1
Reviewer 1 Report
Comments and Suggestions for Authors
The authors conducted a review of the taxonomic status of certain species of the genus Hoplolaimus, highlighting historical nomenclatural issues. This is a very much needed work as the genus is taxonomically complex and carries nomenclatural problems.
The authors carried out an excellent job on sequencing specimens from multiple localities and characterising their morphological traits through detailed imaging. However, the manuscript did not follow the journal's structure for original papers, the aims are not clearly stated, and the full experimental details were not provided. Please, find below my summarised main comments, also stated in the attached document, where other minor comments are included.
Results and Discussion:
These sections must be separated into two. As it is now, it does not follow the journal's guidelines and makes the reading less straightforward. I strongly suggest this should be rewritten following carefully this and other journal requirements (eg., font size in figures) before the next round of reviews.
The new species and the more complete characterisation of the other ones should belong to a Taxonomic section itself, where the new species is properly described following the ICN. Moreover, a taxonomic key or a key character table is recommended to easily differentiate between at least the closest species to the newly described species.
Materials and Methods:
Three different regions were sequenced, with the three relevant trees. However, almost no information about how the trees were generated is shown. Only a reference to a previous first-author paper is made, which is not open access. Therefore, the methodology is not replicable. No relevant information about the tree sampling, the % of discarded trees, or whether the files were compared for estimate convergence is mentioned in the manuscript.
These three sequenced regions are not combined and analysed together. This is commonly used to check for incongruence between regions and to strengthen the topological support in a unique figure. Doing such analysis will direct the reader's attention towards one congruent tree, where phylogenetic relationships are clear. The previous figures could be shown in the Appendix/Supplementary Material.
Moreover, the biogeography section seems rather suggestive without any analysis to support the discussion. Such analysis could be easily done using software like BioGeoBEARS on the concatenated best tree.
I sincerely congratulate the authors on their work on this genus and on finding so many specimens relevant to science.

Author Response
We appreciate all corrections and comments made by referees.
Reply to comments:
Referee 1
1. These sections must be separated into two.
Reply: Results and Discussion section are separated.
2. Taxonomic section itself, where the new species is properly described following the ICN.
Moreover, a taxonomic key or a key character table is recommended to easily differentiate
between at least the closest species to the newly described species.
Reply: Diagnostic keys are recently published in the book: Ghaderi, R.; Hosseinvand, M.;
Eskandari, A. Systematics of the Genus Hoplolaimus (Nematoda: Hoplolaimidae). Scholars’
Press. 2020. 164 pp. We do not see any need in copying this key for this genus and modifying it
by including only a new species similar to many others. Three different regions were sequenced,
with the three relevant trees. However, almost no information about how the trees were generated
is shown. Only a reference to a previous first-author paper is made, which is not open access.
Therefore, the methodology is not replicable. No relevant information about the tree sampling,
the % of discarded trees, or whether the files were compared for estimate convergence is
mentioned in the manuscript
Reply: More information is added to corresponding section of Materials and Methods.
3. Moreover, the biogeography section seems rather suggestive without any analysis to support the
discussion. Such analysis could be easily done using software like BioGeoBEARS on the
concatenated best tree.
Reply: We did NOT have intention to use any phylogeographic software and formulate any
hypothesis in this study. We wanted to provide only maps with locations of several species in the
USA verified by different molecular methods. We would like to underline that there was no such
distribution map created before our study. Several studies already showed that reliable results in
nematode phylogeography could be obtained with analyses of global distribution of samples.
Unfortunately, many regions including South America and Asia with Hoplolaimus are still not
properly sampled, and this limiting sampling dataset could lead to incorrect phylogeographical
conclusions.
4. Please, standardize the way of referring to the author+year when mentioning species/genera for
the first time. The ICZN established to mention without a comma, so please follow this use along
the manuscript.
Reply: Authors names for species are checked and corrected by removing commas.
5. USA. Although commonly used, please indicate the full name in the first use.
Reply:. USA is already used in the title, and we keep it for the Technical Editor decision,
As some of them are synonyms nowadays, please change to "names" rather than taxon (i.e., species)
Reply: In this context we deal with taxa not names.
6. This paragraph fit better into the first, sample description lines from the Materials & Methods section.
Reply: We believe that it would be better to provide readers with more information about why and what we did in this Introduction section.
7. Please, rephrase to indicate background for this study (eg., areas unsampled from previous studies)
Reply: We added a sentence with the main aim of this study, and we keep extended versions for objectives to understand better what we intended to do in our research.
8. "sp. nov." is standardly used across taxonomy, please change all the "sp. n." .
Reply: Both sp. nov. and sp. n. are used to indicate newly described species in scientific literature, and they are essentially interchangeable. Zoological code allows both versions. We prefer shorter version.
9. Please rephrase- Maybe as "Some species were unidentified (A, B1, B2, 1 in Table 1)"
Reply. Corrected.
10. Is this forthcoming?
Reply: ZooBank code for a new species is added
11. Up to here (including this 2.5 paragraph), it all should be together under a "2.1. Taxonomic remarks" or similar type of section.
Reply: All other Hoplolaimus species descriptions should be in separate sections.
12. Do you mean "from Mwesige et al. 2025 (Uganda)" ? In any case, why is it not cited as the rest?
Reply. Full author names for this species are added.
13. Why only independent regions tree are shown? Why not running trees for the concatenated as well to get a more strong idea of the phylogenetic relationships? Were the independent regions incongruent among them?
Reply: New section - 2.6.4. Combining rRNA and COI gene alignment with description of phylogenetic relationships and a tree are added in the manuscript text.
14. Scanning electron microscope (SEM) imaging.
Reply: Corrected.
15. Please, add a Table in the Supp Material that explain each character measured. As it is now, it is impossible to know what Table 1A means (eg., "a", "b", "c", "V%").
Reply: Added.
Reviewer 2 Report
Comments and Suggestions for Authors
This manuscript presents a comprehensive taxonomic revision and molecular phylogenetic analysis of lance nematodes (Hoplolaimus spp.), integrating morphological, morphometric, and multi-locus molecular data (D2-D3 of 28S rRNA, ITS rRNA, and COI). The study convincingly describes a new species, H. floridensis sp. n., resolves long-standing taxonomic ambiguities (e.g., synonymization of H. stephanus with H. galeatus, and H. columbus and H. indicus with H. seinhorsti), and provides biogeographic insights into the genus in North America. The research is methodologically robust, data-rich, and addresses significant gaps in the systematics of agriculturally important plant-parasitic nematodes. The findings have implications for nematode diagnostics, quarantine measures, and pest management.
This manuscript significantly advances the systematics of Hoplolaimus by integrating classical and modern approaches. The taxonomic revisions are well-reasoned, the molecular data are substantial, and the biogeographical synthesis adds ecological depth. Minor revisions have been made to improve clarity and completeness, making this work suitable for publication in the International Journal of Molecular Sciences. Since the authors did not provide line numbers, I have marked the required revisions directly in the PDF file. Please see the notes for details.

Author Response
We appreciate all corrections and comments made by referees.
Referee 2
Citation Format
Reply: Citation format is checked. Author names after species names are not references.
Is there a corresponding reference here?
Reply: Added
I don't understand the meaning of this sentence. Could you please rephrase it based on the content.
Reply: It is taxonomic practice to indicate synonyms or not valid species name after a nematode species name used as the subtitle in a description.
Figure 3, 4, 5: Ensure all clades are clearly labeled, especially species delimitations (e.g., Hoplolaimus sp. B1/B2). Consider adding bootstrap/posterior probability values to all major branches.
Reply: posterior probability values more than 70 are given to all major clades.
The discussion is comprehensive overall, but a concluding summary is still needed.
Reply: Discussion is given in section 3.7. The concluding sentence follows Section 3.8.
Increase font for species names
Reply: Font is increased in all figures
Reviewer 3 Report
Comments and Suggestions for Authors
The manuscript presents a significant contribution to the field of nematology. The research is well structured and application of topotype material provides a strong foundation for this research work. The authors also provided phylogenetic relationships along with geographical distribution for characterization of topotype materials. The authors have used a comprehensive approach for combining morphological and multi-gene molecular data to answer complex taxonomic questions. Due to the comprehensive description of all aspects I think this research work is novel and significant addition to journal.
Author Response
We appreciate all corrections and comments made by referees.
Round 2
Reviewer 1 Report
Comments and Suggestions for Authors
The authors have included all the comments or detailed explained the reasons otherwise.
Congratulations again on their work.